# Genome Sequencing and Analysis Reveal Potential High-Valued Metabolites Synthesized by *Lasiodiplodia iranensis* DWH-2

**DOI:** 10.3390/jof9050522

**Published:** 2023-04-28

**Authors:** Ruiying Li, Pu Zheng, Xingyun Sun, Wenhua Dong, Ziqiang Shen, Pengcheng Chen, Dan Wu

**Affiliations:** The Key Laboratory of Industrial Biotechnology, School of Biotechnology, Jiangnan University, Ministry of Education, Wuxi 214122, China

**Keywords:** *Lasiodiplodia*, genome sequencing, secondary metabolite, 1,3-β-glucan, jasmonic acid

## Abstract

*Lasiodiplodia* sp. is a typical opportunistic plant pathogen, which can also be classified as an endophytic fungus. In this study, the genome of a jasmonic-acid-producing *Lasiodiplodia iranensis* DWH-2 was sequenced and analyzed to understand its application value. The results showed that the *L. iranensis* DWH-2 genome was 43.01 Mb in size with a GC content of 54.82%. A total of 11,224 coding genes were predicted, among which 4776 genes were annotated based on Gene Ontology. Furthermore, the core genes involved in the pathogenicity of the genus *Lasiodiplodia* were determined for the first time based on pathogen–host interactions. Eight Carbohydrate-Active enzymes (CAZymes) genes related to 1,3-β-glucan synthesis were annotated based on the CAZy database and three relatively complete known biosynthetic gene clusters were identified based on the Antibiotics and Secondary Metabolites Analysis Shell database, which were associated with the synthesis of 1,3,6,8-tetrahydroxynaphthalene, dimethylcoprogen, and (R)-melanin. Moreover, eight genes associated with jasmonic acid synthesis were detected in pathways related to lipid metabolism. These findings fill the gap in the genomic data of high jasmonate-producing strains.

## 1. Introduction

*Lasiodiplodia* belongs to the fungal family *Botryosphaeriaceae,* which thrives on a wide range of monocotyledonous, dicotyledonous, and gymnosperm hosts, on woody branches, herbaceous leaves, and grasses, causing die-back and canker diseases in numerous woody hosts [1], such as *Mangifera indica* [2], *Prunus persica* [3], *Saccharum officinarum* [4], and *Vitis vinifera* L. [5]. Currently, *Lasiodiplodia* spp. have been recognized based on conidial morphology combined analysis of ITS and EF1-α [6]. To date, only 19 genomic sequencing data of *Lasiodiplodia* strains are available, with the genome size ranging from 41 to 52 Mb, among which *Lasiodiplodia theobromae* is predominant (Appendix A). Most of the studies on *Lasiodiplodia* spp. have focused on genes that are involved in virulence and pathogenicity. For instance, a set of virulence-related pathogen-associated molecular patterns was determined, and the importance of high temperatures in opportunistic infections was demonstrated based on comparative genome and transcriptome analyses of strains *L. theobromae* CSS-01s, LA-SOL3, and AM2As [7,8,9]. Furthermore, through genomic statistics of *L. gonubiensis* CBS115812, *L. pseudotheobromae* CBS116459, *L. theobromae* CBS164.96, and other draft *Botryosphaeriaceae* genomes, the important role of secreted hydrolases and mycotoxins in the infection of fungal pathogens on their plant hosts was indicated [10]. Moreover, the variations in evolutionary traits and pathogenicity-related genes of *Botryosphaeriaceae* were explored using comparative genomics analysis [11]. However, recently, some active compounds produced by *Lasiodiplodia* spp., such as β-d-glucan and jasmonic acid (JA), as well as their derivatives, have also been investigated. For example, lasiodiplodan, a (1-6)-β-d-glucan with a molecular weight of more than 1.4 × 10^6^ Da and a triple helix structure, was detected in the culture medium of *L. theobromae* [12], which showed potential applications in the medical field owing to its anticoagulant, antimicrobial, antiviral, and antitumor effects [13,14]. In vivo experiments in rats confirmed that lasiodiplodan can be used for the treatment of neurotoxicity and can provide significant protection to the central nervous system [15]. Apart from (1-6)-β-d-glucan, glucan with (1-3)-β linkages in the main chain has also been detected in the culture broth of *L. theobromae* [16], which is considered to be the key active component of various edible and medicinal fungi [17,18]. It has been confirmed that (1-3)-β-glucan-fed mice exhibited improved cognition and enhanced memory [19]. Moreover, the anticervical cancer effect of (1-3)-β-glucan has been verified in vitro [20]. In addition to its therapeutic effect, (1-3)-β-glucan has also received a lot of attention as a potential medical component. For instance, as a carrier of targeted medicine against Her2+ break tumors, (1-3)-β-glucan has been applied to develop targeted delivery system [21]. An anti-mannheimiosis agent, aldsulfin, was isolated from the culture broth of *Lasiodiplodia pseudotheobromae* FKI-4499000, which exhibited antibacterial activity against important livestock pathogens, *Mannheimia haemolytica* and *Pasteurella multocida* [22]. Melanin, a microbial dye, secreted by *L. theobromae* was found to be an environment-friendly pigment for wood dyeing [23]. Furthermore, phytotoxic metabolites extracted from submerged fermentation of *L. pseudotheobromae* have been observed to exhibit bioherbicidal properties [24]. The phytopathogenic fungus, *L. theobromae,* has been reported to produce JA [25], a phytohormone that can disrupt plant defense and invade the host, thus attracting increasing attention as a JA producer [26,27]. JA was first identified and extracted from essential oils obtained from Frangipani (Demole. E et al., 1962) and its derivative, methylated jasmonic acid, became a valuable flavor additive widely used in decorative cosmetics, fine fragrances, shampoos, toilet soaps, household cleaners, and detergents [28]. In addition, JA may have potential pharmaceutical applications, such as in the suppression of human neuroblastoma cells [29], lymphoblastic leukemia, and human prostate, breast, and melanoma cancer cells [30,31]. In a submerged fermentation of *L. theobromae* 2334 [26], a maximum JA concentration of 1.25 g·L^−1^ has been achieved by using potassium nitrate, instead of ammonium nitrate, as a nitrogen source [25].

Studies on *L. theobromae,* a commercial JA-producing species, had mainly focused on the fermentation condition and enzyme activities related to JA synthesized by the mycelium [32,33], and the mechanism of JA production has remained unexplored. In our previous study, a JA-producing fungus named DWH-2 (China Center for Type Culture Collection (CCTCC) strain No. M2017288) was screened and identified as *Lasiodiplodia iranensis* by phylogenetic analysis based on ITS-rDNA gene comparison (Appendix A). This novel JA-producing strain could generate JA of up to 1.5 g·L^−1^ in solid-state fermentation [34]. In the present study, the draft genome sequence of *L. iranensis* DWH-2 was obtained by genome sequencing (SAMN15005723) and compared with Pathogen–Host Interactions (PHI-base), Carbohydrate-Active enzymes (CAZy), Kyoto Encyclopedia of Genes and Genomes (KEGG), and Antibiotics and Secondary Metabolites Analysis Shell (antiSMASH) databases to fully understand the basic composition of the genome and annotate the secondary metabolite biosynthetic gene clusters (BGCs) of the strain. Moreover, genes related to the JA synthesis pathway of *L. iranensis* DWH-2 were identified, thus providing a direction for research on the mechanism of JA synthesis by *L. iranensis*.

## 2. Materials and Methods

### 2.1. Microorganism and Fermentation

*L. iranensis* DWH-2 isolated from *Arachis hypogaea Linn* had been preserved at the China General Microbiological Culture Collection Center (CGMCC strain No. M2017288). The strain was incubated in potato dextrose agar (PDA) slants for 10 days at 28 °C.

### 2.2. Isolation of L. iranensis DNA

The mycelia of *L. iranensis* were incubated for 7 days and then collected, washed with sterile water, dried at 28 °C, and ground in liquid nitrogen. Subsequently, 30 mg of the mycelium powder was used for DNA extraction using a column fungal genomic DNA extraction kit (Sangon Biotech, Shanghai, China), according to the manufacturer’s instructions. The cells were lysed using FPCB solution and β-mercaptoethanol, and chloroform was added after centrifugation to remove impurities. Then, the supernatant was mixed with RNaseA to remove the RNA content, and half volume of BD buffer and half volume of anhydrous ethanol were added to the mixture and the mixture was transferred to the adsorption column. After washing the adsorption column with wash solution and drying, the DNA was extracted with sterilized deionized water, and its concentration was measured. The extracted genomic DNA was detected by Qubit 2.0 and 0.8% agarose gel electrophoresis. The total amount of genomic DNA was 5.23 μg and A260/280 was 1.82. Electrophoresis showed that the genomic DNA had only one band larger than 20 kb and exhibited no degradation and, hence, could be used for library construction and genome sequencing. 

### 2.3. Genome Sequencing, Assembly, and Prediction

The qualified genomic DNA of *L. iranensis* was sequenced using the Illumina platform at the Bohao Biotechnology Co., Ltd. (Shanghai, China). A 400 bp pair-end DNA library was constructed by fragmenting the genomic DNA, terminal repair, 3′-end A-tailing, ligation linker, and enrichment. The concentration of the constructed library was determined using Qubit^®^ 2.0 Fluorometer, and the library size was assessed by Agilent2100. A cluster was generated on the cBot equipped with the Illumina HiSeq sequencer, according to the corresponding process in the cBot User Guide. The paired-end program was selected, and double-ended sequencing was performed in conformity with Illumina User Guide. Quality value was applied to evaluate the quality of the next-generation sequencing data, followed by genome assembly. The raw reads acquired from the Illumina platform were filtered by removing low-quality reads containing <50% of bases with a quality value of >20, bases with a quality value of <20 at the 3′-end, bases with a length of <45, and bases with adaptor contamination to obtain clean reads that can be used for data analysis. The clean reads were assembled to scaffolds with SPAdes-3.5.0 software. The genes in the assembled genome of *L. iranensis* were predicted using the software Prokka combined with the Swiss-Prot library, and 11,224 predicted genes were extracted. 

### 2.4. Functional Annotation

Gene ontology (GO) classification for predicted genes was accomplished by the BLAST2GO algorithm. Clusters of orthologous groups (COG) annotation for the predicted genes was performed by rpsBLAST, and the best and most unique comparison result obtained under the threshold parameter of E-value cutoff was less than or equal to 1 × 10^−5^. The KEGG annotation was performed using KOBAS software, where the predicted genes were aligned with KEGG Orthology (KO) and KEGG pathway library and enriched under the threshold parameter of E-value cutoff less than or equal to 1 × 10^−5^. The JA-biosynthesis-related genes in the *L. iranensis* genome were mined by searching for JA biosynthesis and α-linolenic acid (α-LA) metabolic pathways in KEGG annotation results and by probing for plant JA-related enzymes in GO and COG annotation results. The HMM model of the database for automated carbohydrate-active enzyme annotation (dbCAN, https://bcb.unl.edu/dbCAN2/ (accessed on 29 December 2020)) was used to analyze the potential functional domain of the protein sequence. The parameters were set to E-value < 1 × 10^−5^ and coverage > 0.35 (recommended by dbCAN). 

Annotation with PHI-base database (http://www.phi-base.org/ (accessed on 20 December 2020)) was accomplished using NCBI BLAST-2.12.0+, with E-value < 1 × 10^−5^, and the comparison results with target and query coverage of > 70% were selected from BLASTP alignment. If multiple comparison results were obtained for the same gene, then the best result was chosen according to the frequency of phenotypes and alignment score. Analysis of secondary metabolite BGCs was performed using the antiSMASH 6.0.1 fungal version (https://fungismash.secondarymetabolites.org/ (accessed on 2 December 2020)).

### 2.5. Characterization of the Orthologous Proteins

The CAZy genes or pathogen–host interaction genes in the genomes of the selected strains were extracted. The Orthovenn2 web platform (https://orthovenn2.bioinfotoolkits.net (accessed on 29 December 2020)) was used to identify the orthologous gene clusters in multiple genomes to determine the core gene family of *Lasiodiplodia* sp. and specific genes of each strain.

## 3. Results

### 3.1. Genome Sequencing and Annotation

The genome of *L. iranensis* DWH-2 was assembled using sequencing data generated by the Illumina platform (Appendix A). The characteristics of the *L. iranensis* DWH-2 genome are summarized in Table 1. Approximately 31.5 million Illumina read pairs or about 220× coverage of the genome was assembled, generating a genome sequence of 43.01 Mb comprising 78 scaffolds with N50 lengths of 1.08 Mb. The assembled genome accounted for 98.4% of the reference genome (43.69 Mb) of the genus *Lasiodiplodia* currently available on the NCBI. In addition, the overall GC content of the *L*. *iranensis* DWH-2 genome was approximately 54.82%, and 11,224 coding genes were predicted, among which 4776 (45.2%) genes were assigned, as revealed by the GO annotation system. The distribution of these predicted genes included 4213, 3821, and 3974 genes in the biological process, cellular component, and molecular function, respectively, all of which were relatively uniform and presented cross-overlap in the three categories (Figure 1a,b). Among the Biological Process GO terms, “oxidation–reduction process” occupied the highest percentage (4.54%), followed by “transmembrane transport” (3.85%), “regulation of transcription, DNA-templated” (3.50%), and “transcription, DNA-templated” (2.55%). The most-annotated Cellular Component GO term was “nucleus” (17.78%), followed by “cytosol” (13.78%) and “cytoplasm” (9.02%). “Zinc ion binding” was the most-enriched Molecular Function GO term (5.95%), followed by “ATP binding” (4.08%) and “DNA binding” (3.69%). A total of 6036 (53%) genes were assigned to the KOG database, whereas 7351 (65.5%) genes were assigned to the Swiss-Prot database, which included the highest number of annotated genes. The KOG annotation showed that the major function group was “general function prediction only” containing 1198 genes (Figure 1c). A total of 2345 (20.9%) genes were mapped to 364 pathways in the KEGG database. The top 15 pathways with the highest number of genes annotated are presented in Figure 1d. Among these pathways, the metabolic pathway contained the highest number of genes (980 genes), followed by the biosynthesis of secondary metabolites (409 genes).

### 3.2. Pathogen–Host Interactions Annotation

PHI-base database is used for the identification and presentation of phenotype information on pathogenicity and effector genes and their host interactions [35]. The present study is the first to compare and annotate the *Lasiodiplodia* spp. genomes with the PHI-base database using BLAST-2.12.0+ to evaluate its pathogenicity. According to NCBI, only three *Lasiodiplodia* genomes, namely, *L. theobromae* AM2As, *L. theobromae* LA_SOL3, and *L. theobromae* CSS_01s, had been annotated. Among them, strains LA_SOL3 and CSS_01s had been isolated from *Vitis vinifera*, while strain AM2As had been isolated from cacao stems with vascular streak die-back (VSD) [10]. In the present study, these three *Lasiodiplodia* genomes were analyzed based on the PHI-base database and compared with the *L. iranensis* DWH-2 genome by OrthoVenn2. As shown in Table 2, the number of annotated genes in the *L. iranensis* DWH-2 genome was 268, which is almost similar to those of *L. theobromae* AM2As, CSS_01s, and LA_SOL3 (263, 257, and 266 genes, respectively), but significantly lower than that of *Fusarium venenatum*, a pathogen of maize [36], suggesting that *L. iranensis* DWH-2 is not highly pathogenic. OrthoVenn2 results (Figure 2a) revealed a total of 231 core gene clusters of *Lasiodiplodia* sp., with 1 cluster and 15 singletons found only in *L. iranensis* DWH-2, demonstrating the differences between *L. iranensis* DWH-2 and *L. theobromae*. Based on these findings, the key genes leading to differences in host and pathogenicity of various *Lasiodiplodia* strains could be predicted.

### 3.3. CAZy Annotation

Based on the CAZy database, 558 genes encoding CAZymes were annotated (Figure 3). These genes included 269 glycoside hydrolases (GHs), 127 auxiliary activities (AAs), 73 glycosyltransferases (GTs), 38 carbohydrate esterases (CEs), 26 polysaccharide lyases (PLs), and 3 carbohydrate-binding modules (CBMs). The AAs were classified as AA1–AA9 and AA11, with most of them belonging to the ligninolytic enzymes families. CEs were mainly distributed in nine families, containing CE1, 2, 4, 5, 8, 9, 12, 15, and 16, which catalyze the hydrolysis of sugar esters, including keratinase and various acetylesterases. CBMs mainly included CBM21, 63, and 67, which presented L-rhamnose, cellulose, and starch-particle-binding activities, respectively. PLs comprised PL1, 3, 4, 9, 26, and 42, including rhamnogalacturonan exolyase, pectate lyase, and pectin lyase. The distribution of GHs was very scattered, with a total of 56 families comprising various GHs and α-glucosyltransferase. These results showed that *L. iranensis* DWH-2 has a complex carbohydrate metabolism with the ability to utilize different carbon sources, including lignin, pectin, and xylan, which allows the strain to infect and colonize plants. Thus, these catabolic enzymes could be used as targets to weaken the pathogenicity of *Lasiodiplodia* strains and develop industrial strains. GTs are involved in carbohydrates synthesis and are distributed across 25 families, comprising glycoester synthetase, glycoprotein synthetase, trehalose synthase, and glucan synthase. A 1,3-β-glucan synthase, a key enzyme in 1,3-β-glucan synthesis, was annotated in *L. iranensis* DWH-2. Glucan is an important component of fungal cytoderm, and β-glucan, especially 1,3-β-glucan, has been noted to present major anticancer and immunomodulatory activities [37,38]. Synthesis of β-1,3-glucan from nucleoside diphosphate is catalyzed by hexokinase or glucokinase, phosphoglucomutase, and UTP-glucose-1-phosphate uridylyltransferase [18]. A total of eight genes related to β-1,3-glucan synthesis were annotated in *L. iranensis* DWH-2 (Figure 4), suggesting that *L. iranensis* DWH-2 may generate β-glucan, similar to *L. theobromae* CCT 3966 that can produce β-glucan exopolysaccharide with a productivity of 0.06 g·L^−1^ h^−1^ [39].

The genomes of *L. theobromae* AM2As, *L. theobromae* LA_SOL3, and *L. theobromae* CSS_01s were also analyzed based on the CAZy database, and 602, 591, and 596 genes were annotated, respectively, with slightly more than 558 genes from the *L. iranensis* DWH-2 genome (Figure 3). Significant reductions were noted in AAs and GHs gene families, and OrthoVenn2 results revealed that the orthologous gene families were further associated with the changes in carbohydrate-active enzymes among different strains of *Lasiodiplodia* sp. (Figure 2b). A total of 524 gene clusters were identified as core gene clusters of *Lasiodiplodia* sp., and 34 gene clusters were found in three strains of *L. theobromae*, but not in *L. iranensis* DWH-2 (Appendix A), which reflected the adaptation of *L. theobromae* and *L. iranensis* to different habitats and substrates. In addition, 1,3-β-glucan synthase involved in the synthesis of 1,3-β-glucan was also detected in *L. theobromae*. Alignment of these protein sequences by the BLASTP tool in NCBI showed >90% identity. 

### 3.4. Analysis of Secondary Metabolite BGCs

A total of 47 secondary metabolite BGCs were predicted by antiSMASH, including 2 β-lactone, 9 terpenes, 20 non-ribosomal peptide synthase (NRPS) or NRPS-like, 9 type Ⅰ polyketide synthase (T1PKS), 2 hybrid T1PKS + NRPS-like, 1 hybrid T1PKS + NRPS, and 1 hybrid T1PKS + terpene BGCs (Appendix A). Only nine BCGs showed homologies with known clusters, of which three BCGs showed 100% similarity with known clusters (Figure 5). Almost 81% BGCs did not match with any known gene clusters, indicating that there are many unknown products to be explored, and that *L. iranensis* DWH-2 has the potential to biosynthesize more novel compounds.

Genes within the region 3.1 (BGC 3.1) displayed significant similarity with 1,3,6,8-tetrahydroxynaphthalene (THN) (Figure 5a) BGC (MIBiG: BGC0001258) from *Glarea lozoyensis* [40]. They included a polyketide synthase gene (g1459) and a zinc-finger RING-type protein downstream (g1461), which could regulate polyketide production (Figure 5a). THN is produced by polyketone synthesis and can be translated into 1,8-dihydroxynaphthalene (DHN). The oxidative polymerization of 1,8-DHN is catalyzed by laccase to form DHN-melanin [41]. DHN-melanin, a widespread black pigment in fungi, is an important fungal component that protects the fungi from abiotic stresses, such as toxic metals and electromagnetic radiation [42], and is used to synthesize porous materials with a remarkable ability for gas and toxin adsorption [43]. 

Furthermore, BGC 4.2 (Figure 5b) of strain M2017288 showed a significant BLAST hit with dimethylcoprogen BGC (MIBiG: BGC0001249) from *Alternaria alternata*. Dimethylcoprogen is a type of hydroxamate siderophore synthesized by nonribosomal peptide synthetase. In *L. iranensis* DWH-2, g1949 showed 46.47% sequence similarity with AaNPS6 (GenBank: JQ973666.1) from *A. alternata*, which is required for siderophore biosynthesis [44]. It has been reported that siderophore supplies the necessary iron to promote the process of fungal infection [45].

G2339 of (R)-melanin BGC (Figure 5c) in region 4.5 showed 60.4% similarity with the BGC0001244 gene from *Parastagonospora nodorum*, which has been confirmed to encode the only enzyme required for the production of (R)-melanin [46]. Fungi are the main producers of (R)-melanin [47], which plays a role in the process of infecting plants. In addition to phytotoxicity, (R)-melanin has been noted to show antifungal, antibacterial, and larvicidal activities. A recent study reported that (R)-melanin extracted from *Aspergillus* sp. SPH2 could be used as an insecticide for vectors, such as hard tick *Hyalomma lusitanicum*, and could be a safe and effective tick control agent [48]. Furthermore, 200 μg·mL^−1^ of (R)-melanin was observed to cause 100% death of *Schistosoma mansoni* adult worms [49].

### 3.5. Analysis of Lipid Metabolites

Lipid is an important component of an organism, constituting the basic structure of the cell membrane and playing a physiological role in energy storage, signal transduction, and metabolism regulation. In particular, lipid signaling plays a complex role in the process of fungal invasion and colonization of the host in pathogenic fungi; however, its mechanism remains unclear. In the *L. iranensis* DWH-2 genome, 15 pathways involved in lipid metabolism were annotated through KEGG (Figure 6a) and were found to contain 168 genes. 

JA is one of the oxylipin signaling molecules that respond to stress in plants, such as herbivore attacks or infections by microbial pathogens. It is a cyclopentanone compound derived from fatty acids and is a phytohormone belonging to the jasmonate family. According to the KEGG pathway, α-LA metabolism starts from the release of α-LA from phospholipids, and the final products are jasmonates, traumatic acid, stearidonic acid, and their derivatives. As shown in Figure 6b, eight genes associated with α-LA metabolism were mined in *L. iranensis* DWH-2, including phospholipase A2 that releases α-LA from lecithin and core enzymes of β-oxidation, acyl-CoA oxidase, and acetyl-CoA acyltransferase 1, which possibly catalyze OPC8:0 CoA to sequentially generate OPC6:0 CoA, OPC4:0 CoA, and JA-CoA, shortening the 8-carbon carboxyl side chain to 2-carbon carboxyl side chain after three rounds of β-oxidation (Appendix A). However, the genes involved in the initial steps of α-LA oxidation, such as those encoding plant-like 13S-lipoxygenase (13-LOX), allene oxide synthase (AOS), and allene oxide cyclase (AOC), were not annotated in *L. iranensis* DWH-2, indicating that L. iranensis DWH-2 may have an independently evolved JA biosynthesis pathway different from that of plants. This observation is also supported by the obvious difference between the detected facial selectivity of cyclopentenone reduction in *L. theobromae* and that in plants [33]. Despite significant inconsistencies, the initial step of JA synthesis has been inferred to be the oxidation of LA. Hence, according to the annotation based on the Swiss-Prot database, five lipoxygenase genes (g1161, g6674, g6443, g8110, and g10509) were annotated to be possibly involved in key nodes in α-LA oxidation; however, they had no similarity with the plant 13-LOX (Figure 7a). Interestingly, in addition to the single protein encoded by g6674, proteins encoded by g1161, g6443, g8110, and g10509 have recently been noted to be more closely related to fungal-derived DOX fusion proteins identified by their conserved motifs alignment in fungi (Figure 7b) [50,51]. Furthermore, 9R-DOX and AOS activities were detected in the JA-producing *L. theobromae* disrupted supernatant, which are considered to be comparable to 13-LOX and AOS activities in plants; however, 9R-DOX and AOS activities have not been genetically annotated in *L. theobromae* [50]. Therefore, further investigation of the mechanism and specific enzymes involved in each biosynthetic step in *Lasiodiplodia* sp. is required. 

## 4. Discussion

While most of the studies on JA-producing strains of *Lasiodiplodia* spp. had only focused on the fermentation process, research on the mechanism of fungal JA synthesis is scarce. Exploration of functional genes using only the genome of *Fusarium* sp., which produce a low yield of JA, has revealed the presence of LOX and other key enzymes of the JA synthesis pathway. In the present study, the JA production mechanism of *Lasiodiplodia* spp. was investigated. First, the genome of *L. iranensis* DWH-2, a high JA-yielding strain, was assembled. Then, the JA synthesis route in the pathway of α-LA metabolism and eight genes in the JA synthesis pathway were identified in *L. iranensis* DWH-2 through KEGG annotation; however, these genes did not include plant-like 13S-LOX catalyzing α-LA oxidation and AOS and AOC involved in the subsequent steps, thus indicating the presence of an independently evolved JA biosynthesis pathway in *L. iranensis* DWH-2 different from that in plants. Hence, future investigations of the key enzymes involved in the JA synthesis pathway in *L. iranensis* should focus on the presence of possible fatty acid peroxidase and P450 family enzymes in the strain’s genome. 

The PHI-base database is dedicated to the identification of pathogen–host interaction genes, and each entry in the PHI-base is experimentally validated to ensure the accuracy of the gene function data. Theoretically, PHI-base data support the control of plant diseases and modification of pathogens and hosts. Although *Lasiodiplodia* sp. is a widely influential plant pathogen and a highly anticipated producer of lasiodiplodan and JA in recent years, exploration of pathogen–host interaction genes in its genome is still limited. Analysis of PHI-base homologs in the genus *Lasiodiplodia* revealed the core genes involved in the pathogenicity of this genus and the specific genes in each strain. The specific genes in *Lasiodiplodia* strains indicated the differences in the host and pathogenicity, thus acting as potential targets for virulence modification of industrial strains. Nevertheless, many unannotated specific pathogen–host interaction genes of *Lasiodiplodia* may still exist, and hence, precise gene function research should be performed in future to supplement PHI-base database with experimental data. 

Furthermore, 558 genes encoding enzymes belonging to various CAZy families were detected, among which eight genes were found to encode enzymes involved in the 1,3-β-glucan synthesis, revealing that *L. iranensis* DWH-2 could be a potential candidate for industrial production of 1,3-β-glucan. However, research on the role of 1,6-β-glucan synthetase in lasiodiplodan biosynthesis is limited. Although the production of (1,3;1,6)-β-glucan by *L. theobromae* MMBJ has been reported [12], there has been a lack of evidence of functional genes. The crude glucan extracted from the fermentation broth of *L. iranensis* DWH-2 confirmed the potential of this strain to produce glucan. There are some antibiotics that are probably synthesized, where three BGCs presenting high similarity with known BGCs were found, which might produce THN, dimethylcoprogen, and (R)-melanin. Moreover, the detection of three known BGCs and the presence of unknown BGCs indicated that many metabolites in L. iranensis DWH-2 remained unexplored. The expression of those BGCs and specific antibiotic production need to be confirmed by transcriptome and metabolomics in further research. These findings provide a more comprehensive understanding of the genome of *L. iranensis* DWH-2 and offer insights for an exhaustive gene function research to achieve extensive utilization of *L. iranensis* DWH-2. Development of simple and efficient genetic manipulation methods is crucial for elucidating gene functions and is the foundation for the construction of industrial strains and virulence control, which will be the focus of our future research.

## 5. Conclusions

As an opportunistic plant pathogen, *L. iranensis* DWH2 contains complex GHs, which play a key role in its infection of colonized plant hosts. PHI-base homologs provide rich pathogenic gene targets, and mutations in the key genes may cause significant loss of virulence of phytopathogenic fungi. Moreover, a deeper and comprehensive understanding of the complex metabolic process is the prerequisite for the production of high-valued metabolites and the basis for the construction of safer and more efficient industrial fermentation strains. This study constructed a draft genome of *L. iranensis* DWH-2 and annotated its genes using several databases to achieve exhaustive comprehension of the strain *L. iranensis* DWH-2. The ability of *L. iranensis* DWH-2 to produce high-valued metabolites was ascertained through the mining of key genes, especially those involved in 1,3-β-glucan and JA synthesis. These findings could fill some of the gaps in our understanding of the genus *Lasiodiplodia*. 

## Figures and Tables

**Figure 1 jof-09-00522-f001:**
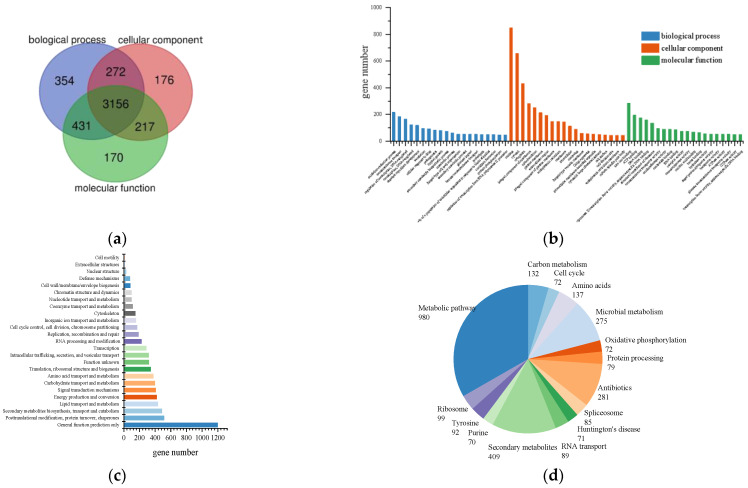
Annotation of predicted genes in *L. iranensis* genome. (**a**) Comparison of the number of genes annotated into three major categories in the GO database, (**b**) GO function classification, (**c**) COG annotation, and (**d**) KEGG pathway annotation.

**Figure 2 jof-09-00522-f002:**
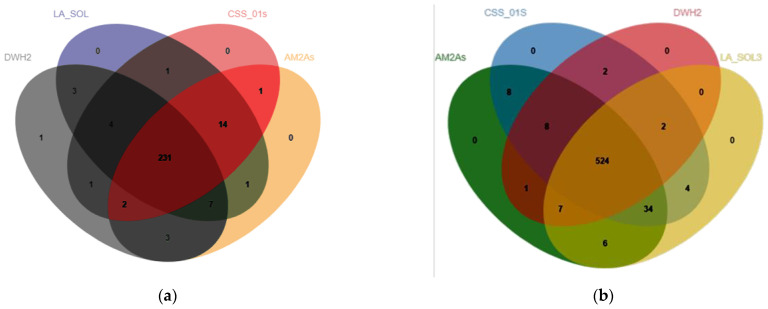
(**a**) Comparison of the PHI-base homologs in the genomes of *L. iranensis* M2017288 and *L. theobromae* AM2As, *L. theobromae* CSS_01s, and *L. theobromae* LA_SOL3 using OrthoVenn2. (**b**) Comparison of CAZymes in the genomes of *L. iranensis* M2017288 and *L. theobromae* AM2As, *L. theobromae* CSS_01s, and *L. theobromae* LA_SOL3.

**Figure 3 jof-09-00522-f003:**
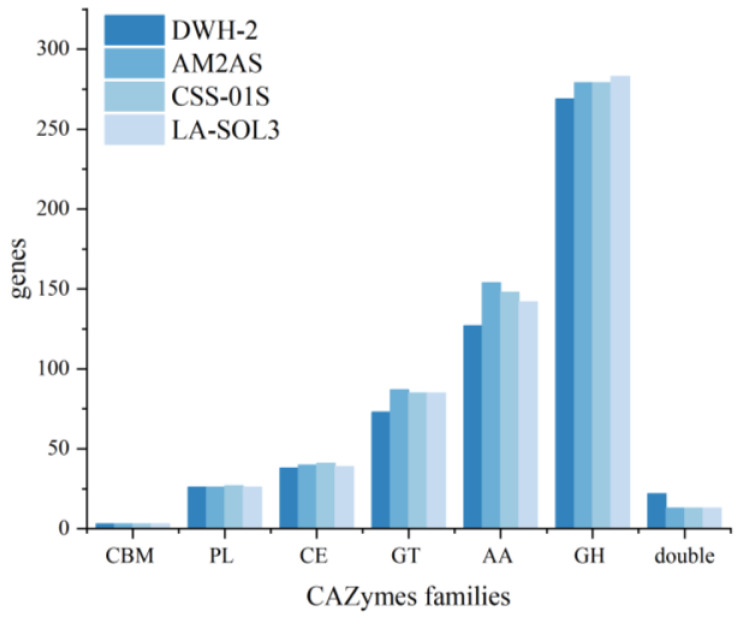
CAZy functional classification and number of corresponding genes in *L. iranensis* DWH-2, *L. theobromae* AM2As, *L. theobromae* CSS_01s, and *L. theobromae* LA_SOL3. CBM, carbohydrate-binding module; PL, polysaccharide lyase; CE, carbohydrate esterase; GT, glycosyltransferase; AA, auxiliary activity; GH, glycoside hydrolase; double, genes containing two domains belonging to different gene families.

**Figure 4 jof-09-00522-f004:**
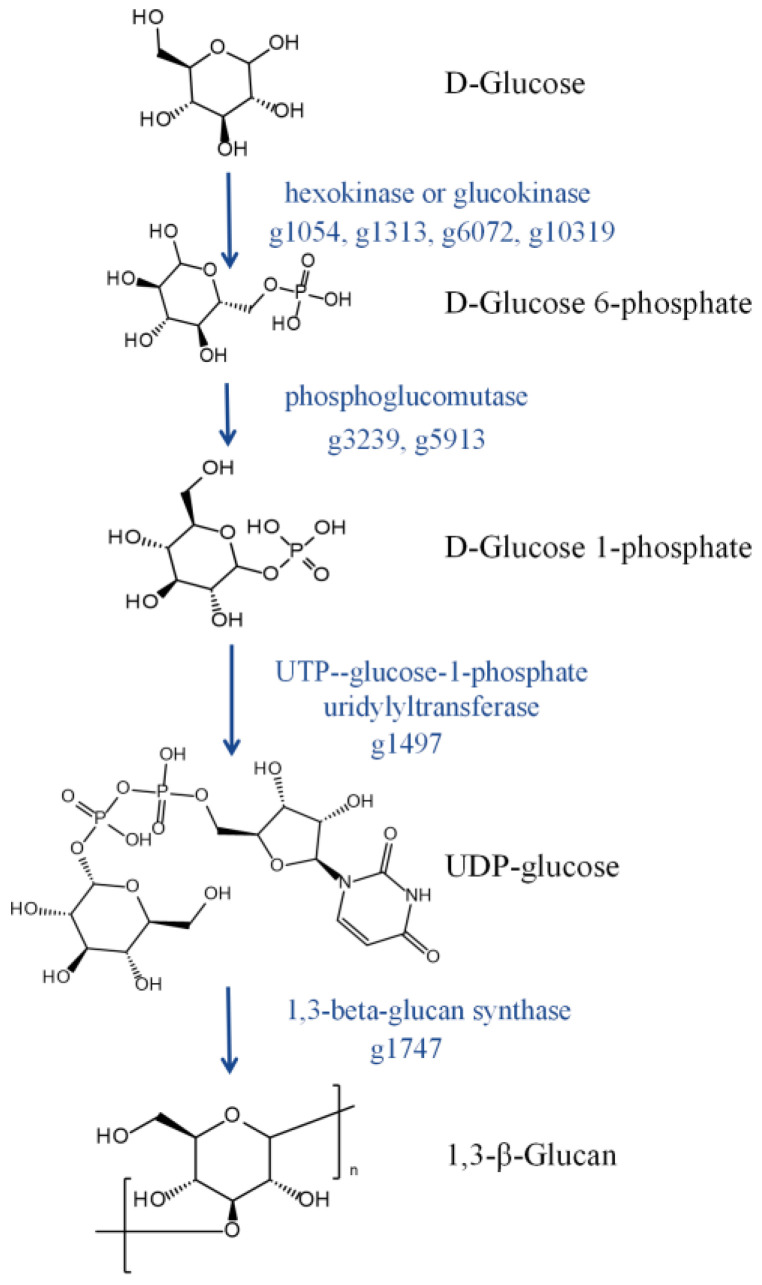
1,3-β-Glucan synthesis pathway. Genes encoding related enzymes are marked on the side of the arrow.

**Figure 5 jof-09-00522-f005:**
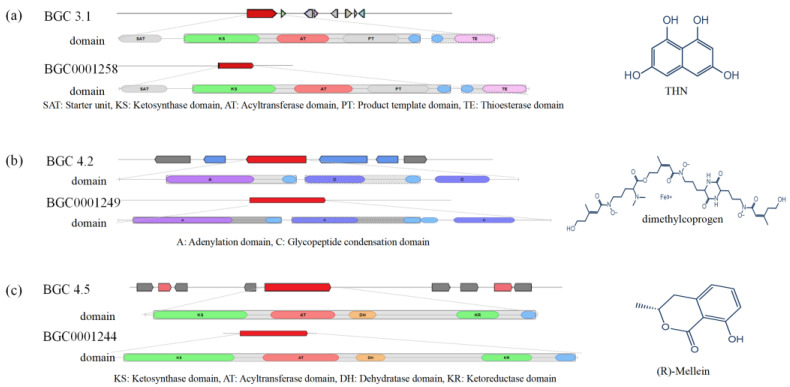
Comparison of BGCs constituents in *L. iranensis* with the identified BGCs involved in the biosynthesis of (**a**) THN, (**b**) dimethylcoprogen, and (**c**) (R)-melanin.

**Figure 6 jof-09-00522-f006:**
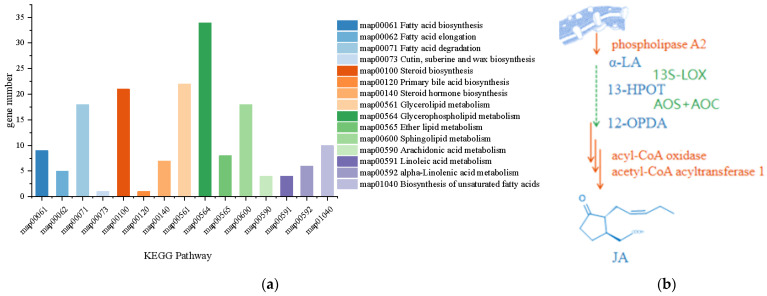
(**a**) Distribution map of lipid metabolism pathways in *L. iranensis*. (**b**) Genes and metabolites involved in the JA synthesis route. Annotated genes were in the orange mark, unannotated genes were in the green mark, and metabolites were in the blue mark.

**Figure 7 jof-09-00522-f007:**
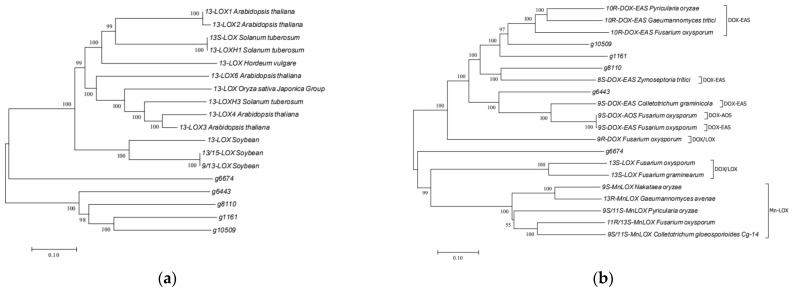
Phylogenetic tree of the mined lipoxygenases aligned with (**a**) 13-LOX in plants and (**b**) fungal dioxygenases.

**Table 1 jof-09-00522-t001:** Characteristics of *L. iranensis* genome.

Features	*L. iranensis*
Size (Mb)	43.01
Sequencing depth	220×
Predicted gene number	11,224
Scaffold number	78
N50 (bp)	1,087,278
Scaffold average length	551,431
Max scaffold length	2,874,908
GC (%)	54.82

**Table 2 jof-09-00522-t002:** PHI-base homologs in the genomes of *L. iranensis* DWH-2, *L. theobromae* AM2As, *L. theobromae* CSS-01s, *L. theobromae* LA-SOL3, and *F. venenatum*.

Phenotype	DWH2	AM2As	CSS-01S	LA-SOL3	*Fusarium venenatum*
Unaffected pathogenicity	72	67	69	70	444
Reduced virulence	141	141	133	141	162
Loss of pathogenicity	23	22	21	22	15
Lethal	27	28	29	28	51
Increased virulence (hypervirulence)	3	3	3	3	7
Chemistry target: resistance to chemical	2	2	2	2	0
Effector	0	0	0	0	0
All	268 ^1^	263	257	266	679

^1^ The identity between query sequence and subject sequence is >70%.

## Data Availability

Data are available on request from the authors.

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
