# Peer review of "Genome Sequencing and Analysis Reveal Potential High-Valued Metabolites Synthesized by Lasiodiplodia iranensis DWH-2"

_jof, 2023, doi:10.3390/jof9050522_

Round 1

Reviewer 1 Report

The first stamen of the authors is that the sequencing analysis reveals the production of high valued metabolites by L iranensis. (1) which are those metabolites? (2) how is their production in comparation with other organisms?

See line 168 Table 1 and line 192 Table 1 (maybe Table 2) see in the text and make correctios if needed.

Line 184 and other places L. iranensis need to be in cursive.

Table 2 Unaffected pathogenicity: what dos means in this case?

Why the authors selected F. venenatum for the comparation

In section 3.4 please defined all the acronyms. (see Table S4)

In the suppletory data please include the description of figure S1 and S2

The paper needs to be revised by an English spiking person.

Avoid using the same words and frasses more the one time in one paragraph.

Author Response

Dear reviewer:

I am very grateful to your comments for the manuscript. According with your advice, we amend the relevant part in manuscript. All of your questions were answered below.

Point 1:The first stamen of the authors is that the sequencing analysis reveals the production of high valued metabolites by L. iranensis. (1) which are those metabolites? (2) how is their production in comparation with other organisms?

Response 1:  Thank you for your kindly comment. (1) We believe that sequencing analysis shows that high-value metabolites synthesized in L. iranensis involve β-glucans, 1,3,6,8-tetrahydroxynaphthalene, dimethylcoprogen, and(R)-melanin in addition to jasmonic acid. (2) The major metabolite jasmonic acid titre was up to 1.5 g·L−1 in solid state fermentation of L.iranensis DWH-2 which was confirmed in our previous study, and it also reported that JA produced by Lasiodiplodia theobromae strain 2334. And β-glucans were potential products of L. iranensis according to reports on the production of β-glucan by fermentation of Lasiodiplodia theobromae and preliminary extraction and identification of glucan from the fermentation broth of L. iranensis. Additionally, the biosynthetic gene clusters of 1,3,6,8-tetrahydroxynaphthalene, dimethylcoprogen, and (R)-melanin, bioactive metabolites demonstrated by recently researches, were found in the genome of L. iranensis.

The word "production" in the manuscript's title was not accurate enough, so we have revised the title to “Genome sequencing and analysis reveal high-value metabolites with potential synthesized by Lasiodiplodia iranensis DWH-2.

Point 2: See line 168 Table 1 and line 192 Table 1 (maybe Table 2) see in the text and make correctios if needed.

Response 2: Thank you very much for your suggestion. We corrected it Table 2 inline 205.

Point 3: Line 184 and other places L. iranensis need to be in cursive.

Response 3: Thank you much for for your suggestion. We have rechecked our manuscript and corrected seven font errors in maintext, located at line 20, 123, 181, 191, 197, 206 and 394. And three errors located in the reference.

Point 4: Table 2 Unaffected pathogenicity: what dos means in this case?

Response 4: Thank you for your suggestion. “Unaffected pathogenicity” is a kind of phenotypes definitied in PHI-base, meaning the transgenic strain which expresses no or reduced levels of a specific gene product(s) has wild-type disease causing ability. Here, we mean the properties related to pathogenic capacity.

Point 5: Why the authors selected F. venenatum for the comparation

Response 5: Thank you for your suggestion. Fusarium and Lasiodiplodia are conditional pathogens belonging to the same genus of Ascomycetes, and the study of pathogen host interaction of Fusarium, especially F.venenatum, was relatively thorough. However, there was no predictive analysis based on PHI base for the genome of Lasiodiplodia sp. Compared with F. venenatum genome, it can help us understand the differences in the number of PHI base homologs between Lasiodiplodia and other pathogenic fungi.

Point 6: In section 3.4 please defined all the acronyms. (see Table S4)

Response 6: Thank you for your suggestion. We have rechecked section 3.4 and Table S4, added full name when abbreviations first appear.

Point 7: In the suppletory data please include the description of figure S1 and S2

Response 7: Thank you for your suggestion. We have added the description of figure S1 and S2 in the supplementary material.

Point 8: The paper needs to be revised by an English spiking person.

Response 8: Thank you for your suggestion. We asked native English speakers to correct the language in the revised manuscript.

Point 9: Avoid using the same words and frasses more the one time in one paragraph.

Response 9: Thank you for your suggestion. We modified this such as line 70,removed “JA was reported”, replace “added” with “mixed” at line 101, and replace “selected” with “chosen” at line 145.

Reviewer 2 Report

In The Ms authors did genome sequencing of Lasiodiplodia iranensis 9 DWH-2 and analysed at various scale.

The study is very interesting and nicely executed with a lot of data.

It can be accepted for publication after suggested changes-

1.      Ms is very poorly written, needs to be thoroughly corrected.

2.      Methods for assembly and annotation should be further elaborated with all the statistical details.

3.      Authors may validate the 1,3-β-Glucan synthesis pathway. By qRT PCR for gene expression.

4.      Discussion is very poorly written. It should be elaborated further.

5.      Conclussion should be rewritten with lacuna in the study and future perspective. 

Author Response

Dear reviewer:

I am very grateful to your comments for the manuscript. According with your advice, we amend the relevant part in manuscript. All of your questions were answered below.

In The Ms authors did genome sequencing of Lasiodiplodia iranensis DWH-2 and analysed at various scale.

The study is very interesting and nicely executed with a lot of data.

It can be accepted for publication after suggested changes.

Point 1: Ms is very poorly written, needs to be thoroughly corrected.

Response 1: We asked native English speakers to correct the language in the revised manuscript.

Point 2: Methods for assembly and annotation should be further elaborated with all the statistical details.

Response 2: Thank you for your suggestion. We have added the details of genome assembly and annotation in line 112-124.

Point 3: Authors may validate the 1,3-β-Glucan synthesis pathway. By qRT-PCR for gene expression.

Response 3: Thank you for your suggestion. We have found that L. iranensis DWH-2. secretes extracellular polysaccharides., and the extracted polysaccharide from the 72-hour fermentation broth of DWH-2 was hydrolyzed by acid. Glucose was detected by HPLC and showed its glucan component was more than 50% (figure 1). The product molecule still needs further identification. We only showed the potential of 1,3-β-Glucan in our MS. Of course, the 1,3-β-Glucan synthesis pathway can be verified by qRT-PCR for gene expression which we need to do in the future.

figure 1. (a) The crude glucan extracted from the fermentation broth of L. iranensis DWH-2 (b) Hydrolyzed product of glucan crude product detected by HPLC confirm to be glucose.(figure can be showed in the file)

Point 4: Discussion is very poorly written. It should be elaborated further.

Response 4: Thank you for your suggestion. We have revised some discussions on the synthesis of jasmonic acid at line 345-351, and added a discussion on PHI annotations at line 359-371 and the application value of synthesizing glucan with L. iranensis DWH-2 at line 375-379. An explanation of future work directions was complemented at the end of the discussion section.

Point 5: Conclusion should be rewritten with lacuna in the study and future perspective.

Response 5: Thank you for your suggestion. We revised the conclusion section to see line 393-399. 

Reviewer 3 Report

The manuscript ‘Genome Sequencing and Analysis reveals Production of High valued Metabolites by Lasiodiplodia iranensis DWH-2’ is an interesting manuscript based on novel findings. The manuscript deals with the discovery of gene clusters and annotation of those genes involved in high value metabolite synthesis by the fungal pathogenic species Lasiodiplodia iranensis DWH-2. The only concern with the manuscript is the little discussion for extensive amount of results.

Authors are suggested to include a brief introduction about the antibiotics produced by the Lasiodiplodia iranensis DWH-2 and their activities in particular THN and dimethylcoprogen.

Figure 1 represents an extensive gene annotation data analysis. However, the results are not discussed well enough.

Materials and methods section is brief and the authors are recommended to include some detailed methodology of the results.

Authors recommends that L. iranensis DWH-2 is suitable for industrial 1,3-β-glucan synthesis. Are there any reports on the production levels of the metabolite from any of the Lasiodiplodia sp? If yes, provide details in the manuscript.

It would be appropriate to provide the detailed steps involved in the JA pathway in figure 6 pointing to the genes that are annotated in the current study as well as the ones that are not annotated and yet to be identified.

Future directions of this research apart from JA pathway analysis could be elaborated in detail.

Include the genome sequence coverage in table 1

Line 25: Lasiodiplodia sp. causes disease in monocotyledons and dicotyledons is a general statement. Can the authors be more specific on which plants they infect more?

Minor issues:

Scientific name L. iranensis is not italicized in some places

The manuscript needs to be proofread to avoid grammar errors.

Line 332: statement doesnt sound right.

Author Response

Dear reviewer:

I am very grateful to your comments for the manuscript. According with your advice, we amend the relevant part in manuscript. All of your questions were answered below.

The manuscript ‘Genome Sequencing and Analysis reveals Production of High valued Metabolites by Lasiodiplodia iranensis DWH-2’is an interesting manuscript based on novel findings. The manuscript deals with the discovery of gene clusters and annotation of those genes involved in high value metabolite synthesis by the fungal pathogenic species Lasiodiplodia iranensis DWH-2. The only concern with the manuscript is the little discussion for extensive amount of results.

Point 1: Authors are suggested to include a brief introduction about the antibiotics produced by the Lasiodiplodia iranensis DWH-2 and their activities inparticular THN and dimethylcoprogen.

Response 1: Thank you very much for your suggestion. Since our current research mainly focused on the de novo sequencing, assembly and annotation of the genomes of L. iranensis DWH-2, and the synthesis ability of metabolites at gene level. THN and dimethylcoprogen were produced by some fungi, and we found the relevant synthetic gene clusters based on BGCs, the experimental validation of antibiotics produced by L. iranensis DWH-2 could be performed in future studies which added in discussion at line 379-385. So, we have revised the title to “Genome sequencing and analysis reveal high-value metabolites with potential synthesized by Lasiodiplodia iranensis DWH-2

Point 2: Figure 1 represents an extensive gene annotation data analysis. However, the results are not discussed well enough.

Response 2: Thank you for your comments and suggestions. We added some discussed in line 166-172.

Point 3: Materials and methods section is brief and the authors are recommended to include some detailed methodology of the results.

Response 3: Thank you for your comments and suggestions. We added some details, as line 112-123.

Point 4: Authors recommends that L. iranensis DWH-2 is suitable for industrial 1,3-β-glucan synthesis. Are there any reports on the production levels of the metabolite from any of the Lasiodiplodia sp? If yes, provide details in the manuscript.

Response 4: Thank you for your comments and suggestions. Some reports provided evidence for the fermentation production of glucan by the Lasiodiplodia sp., and the yield of it reached 0.06 g·L−1h−1 at 72 h. We added this data at line 238.

Point 5: It would be appropriate to provide the detailed steps involved in the JA pathway in figure 6 pointing to the genes that are annotated in the current study as well as the ones that are not annotated and yet to be identified.

Response 5: Thank you for your comments and suggestions. We have revised the figure 6b that point the annotated and unannotated genes.

Point 6: Future directions of this research apart from JA pathway analysis could be elaborated in detail.

Response 6: Thank you for your suggestion. We have revised some discussions on the synthesis of jasmonic acid at line 345-351, and added a discussion on PHI annotations at line 359-371 and the application value of synthesizing glucan with L. iranensis DWH-2 at line 375-379. An explanation of future work directions was complemented at the end of the discussion section.

Point 7: Include the genome sequence coverage in table 1

Response 7: Thanks for your suggestion.  The sequencing coverage of 220× had been added in table 1 at line 181.

Point 8: Line 25: Lasiodiplodia sp. causes disease in monocotyledons and dicotyledons is a general statement. Can the authors be more specific on which plants they infect more?

Response 8: Thanks for your suggestion. We added more specific at line 26-27.

Minor issues:

Point 9: Scientific name L. iranensis is not italicized in some places

Response 9: Thanks for your suggestion. We have rechecked the manuscript and corrected seven font errors in maintext, located at line 20, 123, 181, 191, 197, 206 and 394. And three errors located in the reference, and three errors located in the reference.

Point 10: The manuscript needs to be proofread to avoid grammar errors.

Response 10: Thanks for your suggestion. We asked native English speakers to correct the language in the revised manuscript.

Point 11: Line 332: statement doesnt sound right.

Response 11: Thanks for your suggestion. We revised “ Due to the lack of genomic data, the researches on JA production by Lasiodiplodia spp. were still at the fermentation level, and the research progress on the mechanism of fungal JA synthesis was slow.taking the genome of Fusarium sp. which produce low yield of JA as the research object to explore the LOX and other key enzymes of JA syn-thesis pathway.” to “ While most of the studies on JA-producing strains of Lasiodiplodia spp. had only focused on the fermentation process, research on the mechanism of fungal JA synthesis is scarce. Exploration of functional genes using only the genome of Fusarium sp., which produce low yield of JA, has revealed the presence of LOX and other key enzymes of JA synthesis pathway. In the present study, the JA production mechanism of Lasiodiplodia spp. was investigated. First, the genome of L. iranensis DWH-2, a high-JA-yielding strain, was assembled.”

Round 2

Reviewer 2 Report

Ms is improved now.